# Abrasion and Cavitation Erosion Resistance of Multi-Layer Dip Coated Sol-Gel Coatings on AA2024-T3

**Manasa Hegde** [1,2,*], **Yvonne Kavanagh** [1], **Brendan Duffy** [3] **and Edmond F. Tobin** [1,2,*]

1   Department of Aerospace & Mechanical Engineering, South East Technological University (SETU), Carlow Campus, R93 V960 Carlow, Ireland
2   The Center for Research and Enterprise in Engineering (engCORE), South East Technological University (SETU), Carlow Campus, R93 V960 Carlow, Ireland
3   CREST, Focas Institute, Technological University of Dublin, D07 EWV4 Dublin, Ireland
*   Correspondence: manasa.hegde@itcarlow.ie (M.H.); edmond.tobin@setu.ie (E.F.T.)

**Abstract:** AA2024-T3 are widely used in various applications because of their exceptional physical properties. However, they are susceptible to corrosion and cavitation erosion in aggressive environments due to high concentration of copper. Sol-gel coatings in the field of corrosion prevention are emerging. Improved thickness of coatings significantly improves the barrier effect of the coatings, thereby improving their operational-life in industrial applications. To date, a limited amount of work has been carried out in determining the effect of hybrid sol-gel coatings on abrasion and cavitation erosion of AA2024-T3. The present study investigates the effect of thickness of the coatings on morphology, corrosion, abrasion and cavitation erosion properties of the prepared hybrid sol-gel coatings deposited on AA2024-T3 surfaces. The hybrid sol-gels have been synthesized from 3-trimethoxysilylpropylmethacrylate (MAPTMS), and a zirconium complex prepared from the chelation of zirconium n-propoxide (ZPO), and methacrylic acid (MAAH). AA-2024 T3 were coated using single-dip, double-dip and triple-dip. Abrasion and cavitation erosion tests were performed according to the relevant standards. Structural damage caused by corrosion, abrasion and cavitation erosion was studied by Optical Microscope and Scanning Electron Microscope (SEM). Corrosion protection performance of the coatings was tested using Open Circuit Potential (OCP) and Potentiodynamic polarization (PDS). Results indicated that the multilayer coated samples improved the corrosion, cavitation erosion and abrasion resistance of AA2024-T3. Hence, the prepared silica-based coatings can be proposed as a potential choice for marine renewable energy applications.

**Keywords:** sol-gel coatings; abrasion; cavitation erosion; AA2024-T3

## 1. Introduction

Aluminium alloys are widely used in structural, aerospace and marine applications due to their excellent physical and mechanical properties [1]. However, these materials undergo corrosion and cavitation erosion periodically [1,2]. Cavitation erosion is a complex phenomenon which causes surface degradation of a component [3]. Cavitation erosion is a rapid formation and collapse of bubbles in a liquid caused by large pressure fluctuations [4]. The bubbles collapse near the surface of the metal which results in the generation of shock waves and microjets [5] producing pits, plastic deformation and mass removal of the surface material [6]. Marine components specifically suffer from severe cavitation in the harsh marine environment. Various counter measures are put into practical use to reduce the cavitation, for example the rotational speed of the propellers in ships and boats are lowered [7,8]. Additionally, erosion-corrosion also known as wear corrosion results in the surface degradation of metals [9]. The correlation between erosion and corrosion is an essential subject to understand the interaction between erosion and corrosion. The co-existence of mechanical erosion and electrochemical corrosion is natural in marine environment which is known as tribo-corrosion [10]. Presently, the industries depend on various conditions

namely laboratory testing and use of accelerated erosion testing methods. Numerical analysis was used to study the cavitation erosion effects on various materials. This was challenging as remainder of the flow was incompressible and highly localized variations in density existed between the gas and the liquid [11–13].Various surface treatments namely laser surface alloying, plasma spray, high-velocity oxygen fuel (HVOF) spraying was used to protect the metals from corrosion and cavitation erosion [14,15]. However, technique of spraying gave rise to initial defects such as cracks, pores and incomplete contact interfaces which resulted in early damage of metals under cavitation impact [16]. Along with this, PEO surface treatment producing a protective thick oxide ceramic coating on the metal substrate was used to increase the adhesion of organic coatings, wear and decay resistance on the surface of light metals [17,18]. Despite this fact, there was a formation of pores and cracks in the PEO coatings which were created due to severe thermal stresses and sparkling discharge produced continuously during this process [19]. Additionally, this technique also displayed low coating efficiency and high energy consumption [20]. In the past few years, efforts have been made by the researchers to enhance the erosion resistance of steel by the deposition of protective coatings on the surface [21–23]. TiN coatings on steel was investigated by the authors [24] and they noticed the defects in the coatings leading to reduced incubation time. Furthermore, the coatings resistance of CoMoCrSi against corrosion, oxidation and wear was investigated, but the resistance against cavitation erosion of these coatings was not sufficiently investigated [25]. Hou et.al. [26] conducted a study on the microstructure of Ti-6Al-4V against cavitation erosion and reported that due to repeated bubble collapse, the metal was significantly damaged. In addition to this, hexavalent chromate conversion coatings were used in the protection of aluminium alloys from corrosion, and were eliminated by the environmental regulations as it possessed a serious threat to human health and the environment [27]. Numerous research on cavitation erosion for steel has been reported by researchers [28–30] and only a small selection of work on cavitation erosion for aluminium has been conducted. Taking the above statements into account, the present work is to disseminate the effects of sol-gel coatings on aluminium alloys against corrosion, erosion and abrasion. Organic-inorganic hybrid thin coatings have shown excellent corrosion protection ability for AA2024-T3 alloy [31]. An important feature of sol-gel coatings is their ability to strengthen the adhesion between the metallic substrate and the top coating [32]. Anti-corrosion potential of coatings prepared from 3-trimethoxysilylpropylmethacrylate (MAPTMS) and a zirconium complex of zirconium n propoxide (ZPO) and methacrylic acid (MAAH) in automobile and aerospace applications was studied [33]. Additionally, modification of these coatings with a small amount of cross-linker Hexamethylene diisocyanate (HMDI) and their potential for cavitation erosion and abrasion was also studied [34]. Despite the fact that cavitation erosion of coatings on stainless steel and other metals have been investigated [35], to the best of our knowledge, the abrasion and cavitation erosion behaviour of multilayer thin sol-gel coatings on AA2024-T3 has not been investigated in detail. Therefore, the objective of the present work is to test the abrasion and cavitation erosion resistance of multi-layer dip-coated sol-gel coatings.

## 2. Materials and Methods

### 2.1. Preparation of Sol and Sample Preparation

The coatings are prepared from the mixture of an organosilane precursor MAPTMS (3-methacryloxypropyltrimethoxysilane, Assay 99% in methanol, Sigma Aldrich) and a zirconium complex, prepared from the chelation of zirconium (IV) n-propoxide (ZPO, Assay 70% in propanol, Aldrich) by methacrylic acid (MAAH, $C_4H_6O_4$, Assay >98%, Aldrich). Two different concentrations of Hexamethylene diisocyanate (1% and 1.5%) were added to the prepared coatings. The proportions of the materials employed in these preparations are detailed in Table 1.

**Table 1.** Constituents of the coating.

| Sample Name | Precursors Used | Solvent |
|:-----------:|:---------------:|:-------:|
| SG1 | MAPTMS-ZPO-1% HMDI | |
| SG1.5 | MAPTMS-ZPO-1.5% HMDI | 60% dilution in EtOH |

AA2024-T3 panels were used as the substrate. The substrate was pre-treated with UniClean® 100 series. Films were prepared with the dip-coating process by withdrawal of substrates from the sol at a controlled speed in the range 100 mm/min. Coated substrates were dried in a hot air oven at 120 °C for 1 h [36]. Substrates were coated using one, two and three dip steps, using the same curing conditions. Samples were referred to as SD for single-dip, DD for double-dip and TD for triple-dip.

### 2.2. Surface Characterisation
#### 2.2.1. Corrosion Tests

The corrosion analysis of bare and coated samples was completed with Potentiodynamic scan (Solartron SI 1287/1255B) collected in the region from 1 V to +1 V with a scan rate 10 mV/s at room temperatures with an initial free corrosion potential known as the open circuit potential (5 min dwell time). Each sample was evaluated in sets of 3 replicates. Experiments were performed in 3.5% NaCl solution using conventional three-electrode cell with coated samples as working electrode, Ag/AgCl electrode as reference electrode and Pt electrode as counter electrode coupled with CorrWare2 software to analyse the Tafel graph.

#### 2.2.2. Abrasion Tests

Taber abrader with a pair of CS10 resilient wheels each loaded with a 250 g weight as per ASTMD4060 standards [37] was used to measure the abrasion resistance of the coatings. The abrasion test typically involved 2500 cycles. Abrasion resistance of coatings was assessed by wear index and mass loss. Wear damage was characterised by an optical microscope (Axiolab5, Zeiss, Oberkochen, Germany).

#### 2.2.3. Cavitation Erosion

The cavitation erosion test was conducted up to 36 min using an ultrasonic vibratory cavitation equipment using ASTM G32 standard [38]. Same parameters as described in our previous paper [34] were used. Mass of the bare and coated samples was measured using precision laboratory scales. Cavitation erosion experiments for every sample was carried out at least three times to ensure the repeatability of the experimental results. Cavitation erosion resistance of each sample was evaluated by observing the cumulative mass loss versus exposure time. The dried thickness of the coating was measured (Elcometer 456 Series Digital Coating Thickness Gauge). Morphologies of cavitation surface after the test was observed using SEM (Hitachi SU-70/EDX).

## 3. Results and Discussions
### 3.1. Corrosion Tests

Potentiodynamic polarization scan (PDS) were performed on bare and coated samples to evaluate their anticorrosion properties. Each coating system was evaluated in sets of three replicate panels. Tafel method was applied to determine the coating properties namely corrosion current densities (Icorr) and potential (Ecorr). Polarization resistance (Rp) was calculated using Stern–Geary equation [39].

$$\text{Icorr} = \frac{B}{Rp} \tag{1}$$

where Rp is the polarization resistance and B is a proportionality constant which is calculated B is a constant for the particular system calculated from the slopes of the anodic (βa) and cathodic (βc) from the Tafel region.

$$B = \frac{\beta a \cdot \beta c}{2.3(\beta a + \beta c)} \tag{2}$$

Potentiodynamic polarization curves obtained for single, double and triple coated samples are significantly different from the uncoated sample. From Figure 1, it is observed that the open circuit potential of the coated samples is considerably higher than the bare substrate. The reason for this transition may be because the bare substrate is protected by the developed coatings. Positive shifting of Ecorr indicated the increase of corrosion resistance of the MAPTMS-ZPO coatings. Additionally, corrosion current density is also an important factor to be considered to understand the kinetics of corrosion. From Table 2. It was observed that SG1.5 DD and SG1.5 TD displayed low current densities compared to other samples which signifies that these coatings serve as a physical barrier by hindering the electrochemical process.

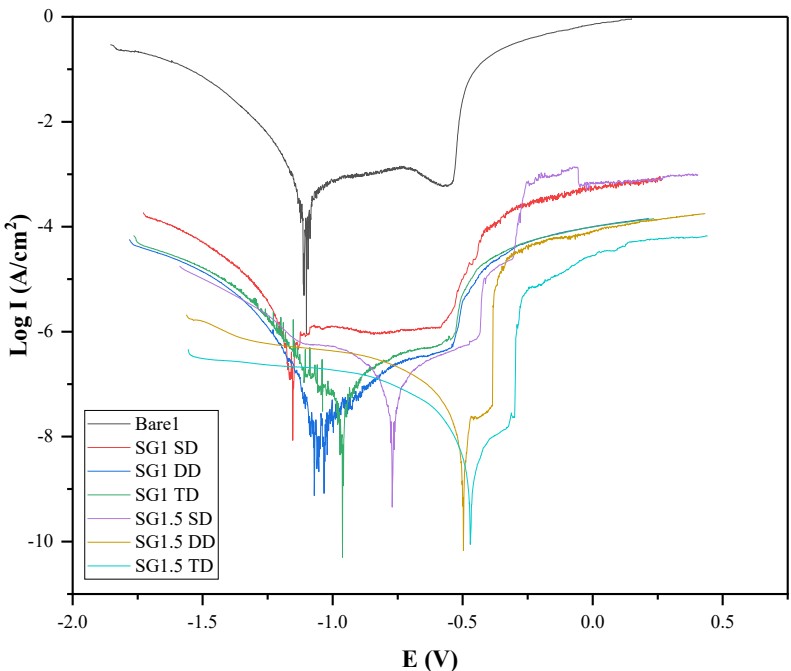

**Figure 1.** Potentiodynamic polarization curves obtained for uncoated, single, double and triple coated samples.

**Table 2.** Electrochemical parameters obtained from Potentiodynamic polarization curves for uncoated, single, double and triple coated samples.

| Sample | Ecorr (V) | Icorr (Acm$^{-2}$) | Rp (Ωcm$^2$) |
|---|---|---|---|
| Bare | −1.26 | $1.36 \times 10^{-5}$ | $1.94 \times 10^{3}$ |
| SG1 SD | −1.20 | $3.93 \times 10^{-5}$ | $6.92 \times 10^{3}$ |
| SG1 DD | −1.17 | $4.68 \times 10^{-7}$ | $6.21 \times 10^{4}$ |
| SG1 TD | −0.54 | $1.06 \times 10^{-6}$ | $1.62 \times 10^{4}$ |
| SG1.5 SD | −1.15 | $9.3 \times 10^{-7}$ | $5.62 \times 10^{4}$ |
| SG1.5 DD | −0.49 | $1.25 \times 10^{-8}$ | $4.27 \times 10^{6}$ |
| SG1.5 TD | −0.51 | $1.7 \times 10^{-8}$ | $1.26 \times 10^{6}$ |

Significant difference was found between one, two and three layers of the coating. Among all the coatings, double and triple coated (with 1.5% HMDI) displayed the highest corrosion resistance.

### 3.2. Abrasion Test

Abrasion resistance was evaluated from mass loss measurements and Taber wear index (TWI). TWI was calculated based on the mass loss data using the expression.

$$TWI = (wa - wb) \times \frac{1000}{N} \tag{3}$$

where wb and wa are the substrate weights before and after the test, N is the number of cycles.

TWI values for coated and uncoated samples are shown in Table 3. Lesser values of TWI signify good abrasion resistance of the coated specimen. From Table 3, it is observed that, double coated substrate with 1.5% of HMDI have lower values of TWI compared to single and triple coated substrate. However, uncoated bare substrate shows higher TWI values.

**Table 3.** Taber Wear Index (TWI) values for uncoated, single, double and triple coated samples.

| Sample | Bare | SG1 SD | SG1 DD | SG1 TD | SG1.5 SD | SG1.5 DD | SG1.5 TD |
|---|---|---|---|---|---|---|---|
| No. of Cycles | | | | TWI | | | |
| 100 | 10 | 0 | 0 | 0 | 0 | 0 | 0 |
| 300 | 11.3 | 18 | 0.6 | 0.6 | 16 | 0.6 | 2.6 |
| 500 | 12 | 8.8 | 5.6 | 4.2 | 4.8 | 0.6 | 1.6 |
| 700 | 9.8 | 7.8 | 1.1 | 4.2 | 8.1 | 0.7 | 2.2 |
| 900 | 8.5 | 5.1 | 3 | 5 | 13.4 | 1.2 | 2.9 |

Wear pattern of the coatings was evaluated by mass loss measurements. Cumulative mass loss was recorded for both uncoated and coated substrate before and after each test. Figure 2 shows the changes in mass loss versus no. of cycles for the uncoated and coated substrates. SG1.5 DD and TD displayed lowest mass loss even after undergoing 2500 cycles. Uncoated and single coated substrate had majority of mass loss comparatively.

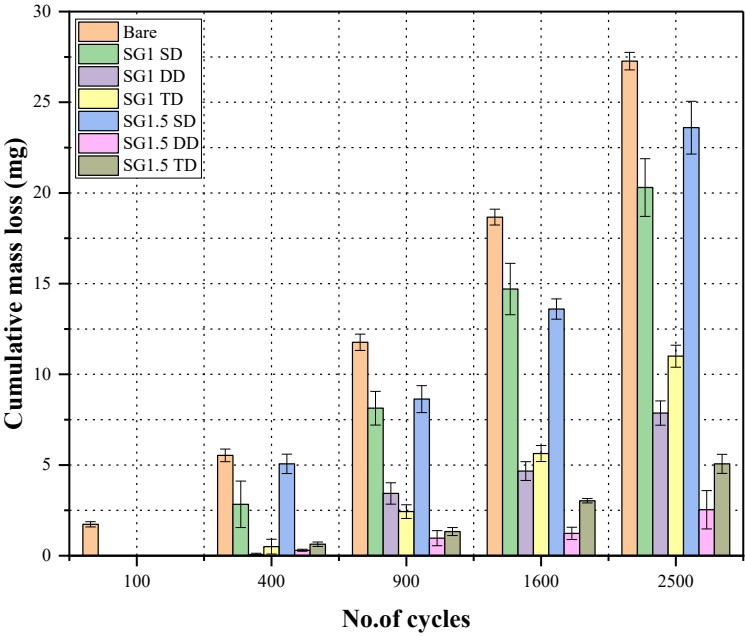

**Figure 2.** Graphical illustration of abrasion test results.

Microscopic studies of the abraded surface after 2500 cycles exhibit that the coatings are damaged, but greater defects were detected in the uncoated and single coated substrates. From Figure 3a,d, a large number of cracks can be observed on the side of abraded surfaces. No major caracks can be seen in Figure 3b,e. Figure 3c,f indicates that no delamination or fragmentation of the coating from the substrate has occurred. Intrinsic abrasion resistance of the coating is rated by the distribution of the wear track caused by Taber test, which means that the abrasion test should neither cause damage to the coating nor the substrate. Abrasion test with 2500 cycles disseminate that uncoated and single-dip coatings are damaged with deeper defects whereas double and triple-dip coatings are less damaged comparatively.

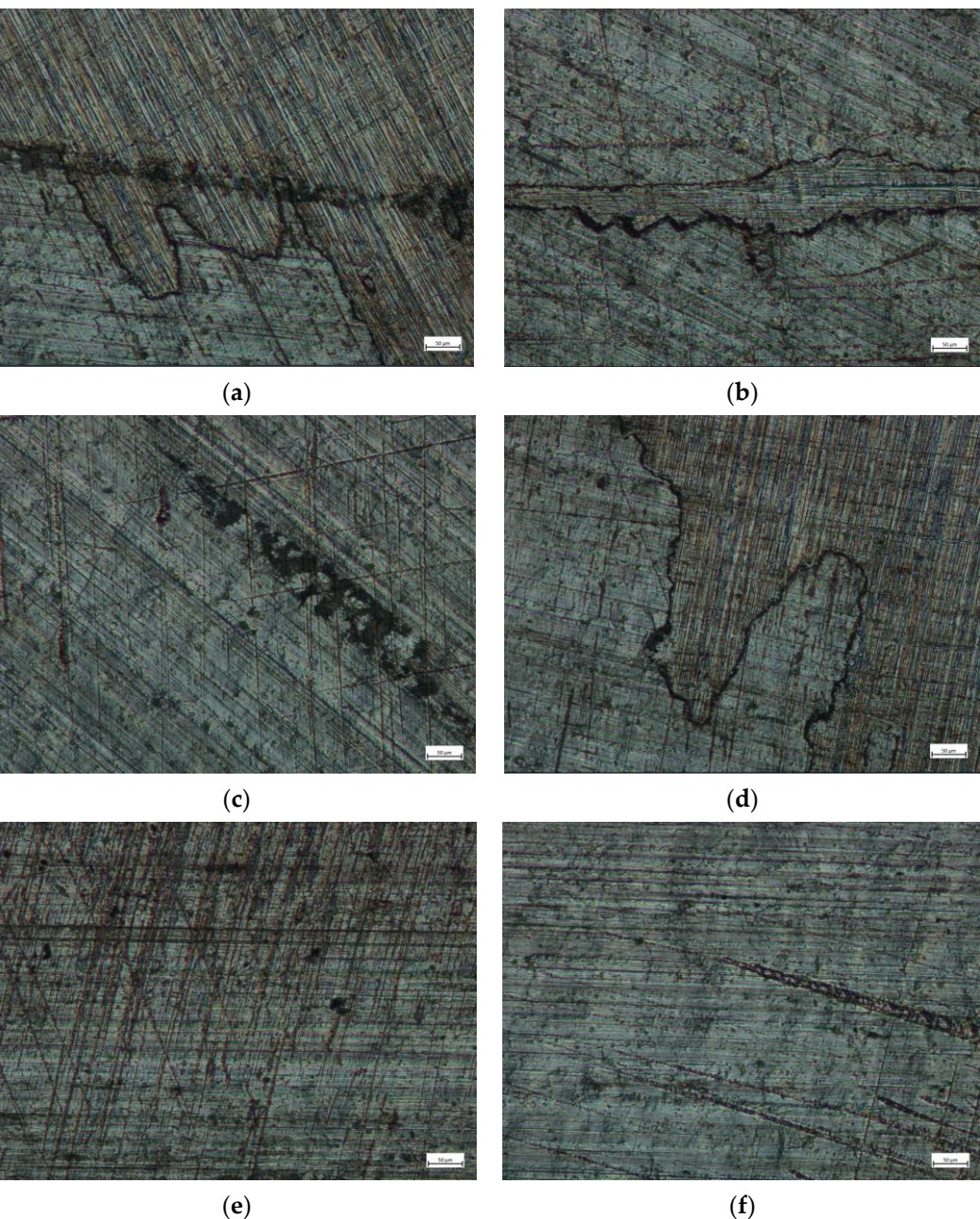

(a)   (b)

(c)   (d)

(e)   (f)

**Figure 3.** *Cont.*

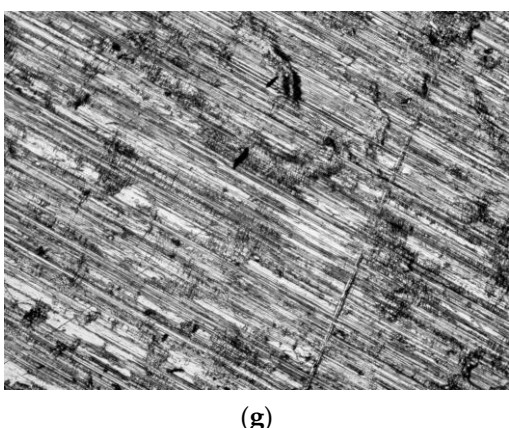

(**g**)

**Figure 3.** Optical micrographs (50 μm) of abraded coatings after 2500 cycles (**a**) SG1 SD, (**b**) SG1 DD, (**c**) SG1 TD, (**d**) SG1.5 SD, (**e**) SG1.5 DD, (**f**) SG1.5 TD and (**g**) bare substrate.

### 3.3. Cavitation Erosion

Accelerated short-period cavitation erosion laboratory tests using an ultrasonic transducer were performed, and a comparative study among the uncoated and coated substrates was conducted. The cavitation erosion resistance of the coatings was tested for relatively short periods of time as the thickness of the coating was very thin (2–6 μm). The aim was to observe at what particular time the coating was delaminated from the metal. Throughout the tests, water quality, temperature, stand-off distance, frequency and amplitude were kept constant. The mass loss of the dried samples measured after each test. The graph in Figure 4 shows the cumulative mass loss of the samples as a function of time. It can be observed that all the samples have linear mass loss which means that the wear rate of the samples are constant during the test. It is also observed that there is no severe mass loss for coated substrates throughout the test. This is due to the incubation period occurring in the beginning during which the metal surface is plastically deformed, whereas for uncoated substrate, there was majority of mass loss since the beginning of the test, wear begins immediately after the first minute of the test. Single coated samples (SG1 and SG1.5) were not very resistant to cavitation as the coating was delaminated after only 1 min of the test. Double and triple-coated samples with 1% HMDI was resistant to cavitation for up to 4 min. However, double and triple-coated samples with 1.5% HMDI was resistant to cavitation up to 9 min. Therefore, it can be observed that thickness influences cavitation erosion. As the thickness of the coating is increased, the cavitation resistance increased by extending the incubation period and reducing mass loss rate.

The surface morphology of the samples after 36 min of cavitation test was studied using SEM. The images shown in Figure 5 are all obtained from severely damaged areas. The thickness of the coatings measured using DFT showed 2, 4 and 6 microns for SD, DD and TD coatings. This was confirmed by SEM while examining delaminated areas. It can be observed that the damage mechanism is different for samples with different thickness. The end of incubation time of the coated samples is approximately after 36 min according to the mass loss curve, single coated SG1 and SG1.5 are completely damaged with rougher surface area which indicates heavy erosion caused by the cavitation test, whereas DD and TD coated substrates exhibits an eroded surface but with smaller damaged craters. Several small craters with diameters of 3.5–4.5 μm were observed (Figure 6) on the eroded surface of SG1.5 DD and TD coated substrates. The craters on the eroded surface are identified by the repeated collapse of bubbles around the pits and pores.

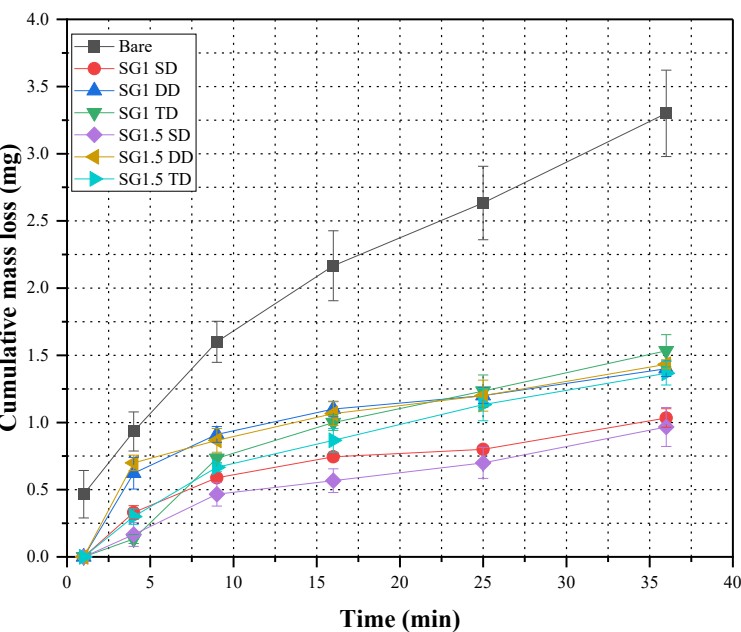

**Figure 4.** Graphical illustration of cumulative mass loss during cavitation test for uncoated, single, double and triple coated samples.

|  | SD | DD | TD |
| --- | --- | --- | --- |
| **SG1** | | | |
| **SG1.5** | | | |
| **SG (un-damaged)** | | | |

**Figure 5.** SEM images of the coatings after and before 36 min cavitation test.

(**a**) (**b**)

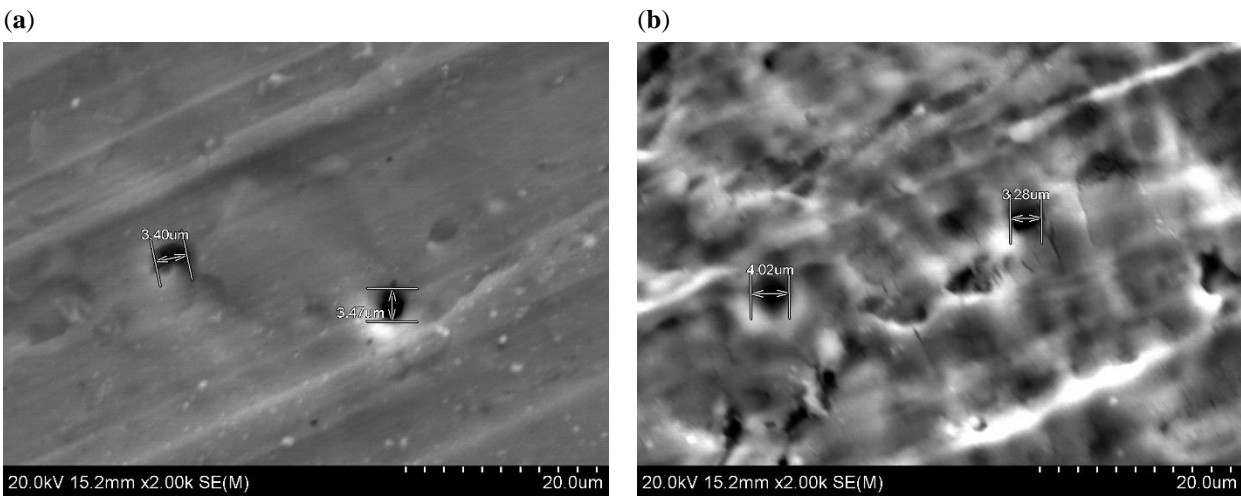

**Figure 6.** SEM images showing diameter of crater formation in (**a**) SG1.5 DD, (**b**) SG1.5 TD.

Regardless of all the damage caused to the coatings by cavitation test, it was observed that some small parts of the coatings remained on SG1.5 DD and SG1.5 TD samples even after 36 min of the test. Figure 7 shows SEM micrographs of thickness of SG1.5 DD and SG1.5 TD after 36 min of cavitation test.

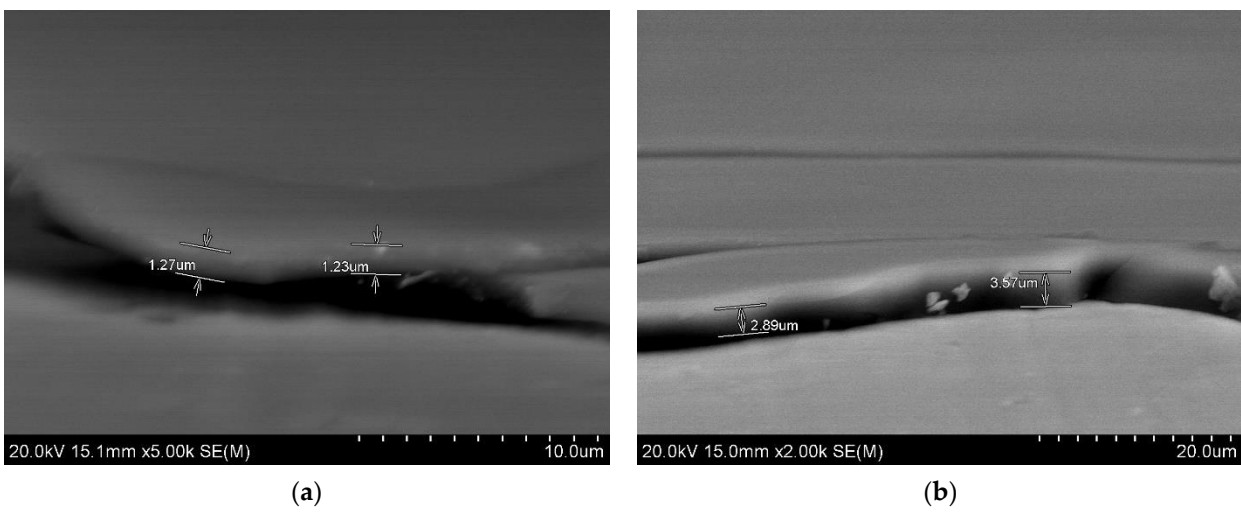

(**a**) (**b**)

**Figure 7.** SEM images showing the thickness of (**a**) SG1.5 DD, (**b**) SG1.5 TD after 36 min of cavitation test.

## 4. Conclusions

The present work shows that the developed sol–gel coatings on AA2024-T3 provide erosion-corrosion protection and can be used as an alternative for chromate-conversion coatings. Active corrosion protection was demonstrated by both double and triple coated substrates through potentiodynamic measurements. Abrasion tests demonstrated satisfactory wear behavior for thicker coatings (double and triple coated). A series of accelerated short-period cavitation erosion tests for a total duration of 36 min showed that the resistance to cavitation erosion of double and triple coated samples was higher than that of the uncoated AA2024-T3. The coating morphology displayed significant wear for single coated samples whereas double and triple coated substrates exhibited an eroded surface with smaller scale erosion pits and craters. Although the prepared coatings seem to improve the corrosion, abrasion and cavitation erosion resistance to a certain level, its role in the marine environment is not fully conceived. Therefore, further investigations are necessary. However, the sol-gel coating process proved to be a promising alternative to other environmentally hazardous coating systems.

**Author Contributions:** M.H.: Conceptualization, Methodology, Formal Analysis, Investigation, Writing-Original draft preparation, Funding acquisition; Y.K.: Supervision; B.D.: Resources, Writing-Reviewing and Editing, Supervision; E.F.T.: Conceptualization, Writing-Reviewing and Editing, Supervision. All authors have read and agreed to the published version of the manuscript.

**Funding:** This work was supported by the Irish Research Council under award number [GOIPG/2021/24].

**Data Availability Statement:** The data presented in this study are available on request from the corresponding author.

**Acknowledgments:** The research is funded by the Irish Research Council Postgraduate fellowship scholarship. I would also like to thank Frederick Odunjo, CREST TU Dublin for his help in Scanning Electron Microscope.

**Conflicts of Interest:** The authors declare no conflict of interest.

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
