# Peer review of "Abrasion and Cavitation Erosion Resistance of Multi-Layer Dip Coated Sol-Gel Coatings on AA2024-T3"

_cmd, doi:10.3390/cmd3040036_

Round 1

Reviewer 1 Report

As described in section 2.1, the deposited layers were dried at 120 degrees for 1 hour. This treatment only causes the evaporation of some solvents (methanol, water...), but it is not a high enough temperature to cause densification and crystallization of the deposited layers. Therefore, in my opinion, these are coatings with poor consistency and protection.

Figure 1 shows SEM images of the thickness of certain layers. However, the thickness indicated in the images corresponds to delaminated areas due to the excess thickness of the layers, therefore, the average thickness of the layers on the surface without delamination must be much lower than indicated. I recommend measuring and indicating the correct thickness of the layers without delamination or cracking.

Author Response

Point 1: As described in section 2.1, the deposited layers were dried at 120 degrees for 1 hour. This treatment only causes the evaporation of some solvents (methanol, water...), but it is not a high enough temperature to cause densification and crystallization of the deposited layers. Therefore, in my opinion, these are coatings with poor consistency and protection.

Response 1: The materials hydrolyse and condense in solution to form a colloidal suspension. The curing of the coating can occur at room temperature as the silane species cross-link. Examples of such coatings are available commercially (e.g. Nanoceramic car coatings, e.g. Gtechniq - CSL Crystal Serum Light - Ceramic Coating). The coatings in this paper are cured at 120C to ensure the thicker coatings fully protect the surface. The team has commercialised other coatings and can confirm the protective capabilities (to an obvious limit).

Point 2: Figure 1 shows SEM images of the thickness of certain layers. However, the thickness indicated in the images corresponds to delaminated areas due to the excess thickness of the layers, therefore, the average thickness of the layers on the surface without delamination must be much lower than indicated. I recommend measuring and indicating the correct thickness of the layers without delamination or cracking.

Response 2: A DFT meter was used to measure the thickness of the coatings (Single coated substrate displayed a thickness of 2 microns, double-coated substrate displayed a thickness of 4 microns and triple-coated substrate displayed a thickness of 6 microns), but would not be able to provide images.

Reviewer 2 Report

The main conceptual problem of the paper by Hegde et al. is that AA2024 must not be used in marine applications. There are other alloys e.g. 5xxx or 6xxx for that purpose. Is AA2024 a model alloy, what was the purpose of using it? To which extend do the coatings provide protection before developing pitting activity? It would be good to compare Taber Wear Index obtained in this work with other works. This will unbiasedly present the drawbacks/benefits of the sol-gel coatings compared to other coatings, e.g. PEO, anodic coatings etc.

Introduction could mention the efficiency of PEO and anodic coatings in protection against wear and scratches.

Nevertheless, the paper is well-written and merits publication with corrections.

2 Materials and methods

Line 85, Give a full name of HMDI

Line 103, what is “saturated Ag colomel electrode” KCl saturated calomel electrode?

Figure 2. Is mass loss normalized, to which area? Why not indicating that on the axis? Same for Figure 4.

Figure 5. SEM images of Undamaged SG coatings are needed, otherwise it is not clear from where does the porosity come from.

Line 222-224 Quote: “Regardless of all the damage caused to the coatings by cavitation test, it was observed 222 that some small parts of the coatings still remained on SG1.5 DD and SG1.5 TD samples 223 even after 36min of the test.” Does this mean that the surface presented in Fig.5 is mostly bare? How much of the bare surface appeared after the tests? It is better to mark delaminated zones on the images.

Line 230, Figure 1 follows Figure 6, why is that?

Author Response

Point 1: The main conceptual problem of the paper by Hegde et al. is that AA2024 must not be used in marine applications. There are other alloys e.g. 5xxx or 6xxx for that purpose. Is AA2024 a model alloy, what was the purpose of using it? To which extend do the coatings provide protection before developing pitting activity? It would be good to compare Taber Wear Index obtained in this work with other works. This will unbiasedly present the drawbacks/benefits of the sol-gel coatings compared to other coatings, e.g. PEO, anodic coatings etc.

Response 1: Yes, The 2024 was used as a model alloy. Similar coatings on marine alloys have been discussed with Mercury Marine, a leading manufacturer of otboard engines.The coatings developed in the present work is very thin (has a thickness of 2-6microns) and we don’t have but we have no direct comparators of Taber wear Index for this work.

Point 2: Introduction could mention the efficiency of PEO and anodic coatings in protection against wear and scratches.

Response 2: Introduction has been improved by the authors.

Point 3: Line 85, Give a full name of HMDI

Response 3: full form of HMDI is mentioned in Line 85.

Point 4: Line 103, what is “saturated Ag colomel electrode” KCl saturated calomel electrode?

Response 4: Ag/AgCl was used as the reference electrode.

Point 5: Figure 2. Is mass loss normalized, to which area? Why not indicating that on the axis? Same for Figure 4.

Response 5: Initial mass loss is of the ultrasonic horn used for the cavitation test. Hence, it's of the same area.

Point 6: Figure 5. SEM images of Undamaged SG coatings are needed, otherwise it is not clear from where does the porosity come from.

Response 6: The SEM images of Undamaged SG coatings have been added.

Point 7: Line 222-224 Quote: “Regardless of all the damage caused to the coatings by cavitation test, it was observed 222 that some small parts of the coatings still remained on SG1.5 DD and SG1.5 TD samples 223 even after 36min of the test.” Does this mean that the surface presented in Fig.5 is mostly bare? How much of the bare surface appeared after the tests? It is better to mark delaminated zones on the images.

Response 7: Figure 5 indicates the SEM images of the part which underwent the cavitation test. The coatings were gone after 36 min of the test. So, the images indicate mostly bare substrate after the delamination of the coatings.

Point 8: Line 230, Figure 1 follows Figure 6, why is that?

Response 8: Apologies, it was a typo. The figure number has been corrected (Figure 7).

Reviewer 3 Report

The aim of the manuscript is interesting. However, some minor improvements must be performed. 

1. Please explain why the potentiodymanic polarization tests were performed from -1V?

2. Please explain how did you perform the erosion tests during 36 min?

3. Please improve the conclusions. Some phrases presented as conclusions sound as methodoloy. 

Author Response

Point 1: Please explain why the potentiodymanic polarization tests were performed from -1V?

Response 1: Different parameters were tested earlier. For the work presented in this manuscript, the value was taken from -1V

Point 2: Please explain how did you perform the erosion tests during 36 min?

Response 2: Cumulative time measurements was taken (1min, 3min, 5min,7min,9min and 11min)

Point 3: Please improve the conclusions. Some phrases presented as conclusions sound as methodology.  

Response 3: Have improvised the conclusions.

Round 2

Reviewer 1 Report

The Manuscript has been improved in several aspects.

Regarding Point 1, the curing temperature may be sufficient to produce a relative protective capability.

Regarding Point 2, I consider it a pity that more reliable layer thickness measurements are not available, only SEM images of delaminated layers.

Author Response

Point 1: the curing temperature may be sufficient to produce a relative protective capability.

Response 1: The authors have added a reference for further clarification.

Point 2, I consider it a pity that more reliable layer thickness measurements are not available, only SEM images of delaminated layers.

Response 2: The thickness of the coatings was measured by a DFT (Elcometer 456 Series Digital Coating Thickness Gauge) which displayed a thickness of approximately 2, 4, and 6 microns ( for SD, DD, and TD coatings), and this was confirmed by SEM while examining delaminated areas.

Reviewer 2 Report

More revisions...

line 56 remove double " [[ "

line 78 typo: must be "Hexamethylene diisocyanate (HMDI)"

line 121 typo: must be "..paper [34] were used.."

Line 236-237: "In this work, we have shown that the sol-gel technology contributes to the synthesize of multifunctional coating systems for anticorrosion, anti-erosion and wear properties. " I cannot understand this sentence, please revise it.

Line 240: "EIS measurements" Were there any EIS measurements in this work? Maybe authors mean PD / Tafel measurements?

Author Response

Point 1: line 56 remove double " [[ "

Response 1: Corrected.

Point 2: line 78 typo: must be "Hexamethylene diisocyanate (HMDI)"

Response 2: Corrected.

Point 3: line 121 typo: must be "..paper [34] were used.."

Response 3: Corrected.

Point 4: Line 236-237: "In this work, we have shown that the sol-gel technology contributes to the synthesize of multifunctional coating systems for anticorrosion, anti-erosion and wear properties. " I cannot understand this sentence, please revise it.

Response 4: Revised

Point 5: Line 240: "EIS measurements" Were there any EIS measurements in this work? Maybe authors mean PD / Tafel measurements?

Response 5: Corrected.

Reviewer 3 Report

The manuscript was revised and improved. It can be accepted for publication in the present form.

Author Response

Thank you so much for your time in reviewing our manuscript.